# Modulation of the Immune Response by Deferasirox in Myelodysplastic Syndrome Patients

**DOI:** 10.3390/ph14010041

**Published:** 2021-01-07

**Authors:** Hana Votavova, Zuzana Urbanova, David Kundrat, Michaela Dostalova Merkerova, Martin Vostry, Monika Hruba, Jaroslav Cermak, Monika Belickova

**Affiliations:** 1Department of Genomics, Institute of Hematology and Blood Transfusion, U Nemocnice 1, 128 20 Prague, Czech Republic; hana.votavova@uhkt.cz (H.V.); david.kundrat@uhkt.cz (D.K.); michaela.merkerova@uhkt.cz (M.D.M.); martin.vostry@uhkt.cz (M.V.); monika.hruba@uhkt.cz (M.H.); 2First Faculty of Medicine, Charles University, Katerinská 32, 121 08 Prague, Czech Republic; zuzana.urbanova@uhkt.cz; 3Department of Clinical Hematology, Institute of Hematology and Blood Transfusion, U Nemocnice 1, 128 20 Prague, Czech Republic; jaroslav.cermak@uhkt.cz

**Keywords:** myelodysplastic syndrome, iron chelation, deferasirox, molecular mechanisms

## Abstract

Deferasirox (DFX) is an oral iron chelator used to reduce iron overload (IO) caused by frequent blood cell transfusions in anemic myelodysplastic syndrome (MDS) patients. To study the molecular mechanisms by which DFX improves outcome in MDS, we analyzed the global gene expression in untreated MDS patients and those who were given DFX treatment. The gene expression profiles of bone marrow CD34^+^ cells were assessed by whole-genome microarrays. Initially, differentially expressed genes (DEGs) were determined between patients with normal ferritin levels and those with IO to address the effect of excessive iron on cellular pathways. These DEGs were annotated to Gene Ontology terms associated with cell cycle, apoptosis, adaptive immune response and protein folding and were enriched in cancer-related pathways. The deregulation of multiple cancer pathways in iron-overloaded patients suggests that IO is a cofactor favoring the progression of MDS. The DEGs between patients with IO and those treated with DFX were involved predominantly in biological processes related to the immune response and inflammation. These data indicate DFX modulates the immune response mainly via neutrophil-related genes. Suppression of negative regulators of blood cell differentiation essential for cell maturation and upregulation of heme metabolism observed in DFX-treated patients may contribute to the hematopoietic improvement.

## 1. Introduction

Myelodysplastic syndrome (MDS) is a heterogeneous group of clonal stem cell disorders characterized by ineffective hematopoiesis and a higher risk of evolution to acute myeloid leukemia (AML). Approximately 60–80% of MDS patients have symptomatic anemia, and the majority of these patients need repeated red blood cell (RBC) transfusions as supportive care [1]. Long-term exposure to RBC transfusions leads to the iron accumulation that has detrimental effects on hematopoiesis, especially on erythropoiesis. The toxic effect of iron overload (IO) is mediated by the generation of reactive oxygen species (ROS) that increase oxidative stress. It causes DNA oxidation, mitochondrial damage and the peroxidation of membrane lipids, resulting in tissue damage.

MDS patients with IO have a decreased life expectancy due to cardiac and liver failure and need to be treated with iron chelation therapy (ICT). The clinical utility of ICT in MDS patients with RBC transfusion dependency is still under debate. Clinical benefits of ICT such as decreased incidence of cardiac events, diabetes and hepatic impairment; reduced bacterial and fungal infections; improvements in hematopoiesis; lower risk of leukemic transformation; and improved outcome after allogeneic stem cell transplantation have been observed in various subgroups of MDS patients [2]. Liu et al. performed a meta-analysis of 14 studies focused on beneficial effects of ICT in MDS and concluded that the ICT is associated with both prolonged overall survival and leukemia-free survival [3].

In recent decades, various iron chelators have been developed, such as deferoxamine (DFO), deferasirox (DFX) and deferiprone (DFP). The most commonly used iron chelator in MDS is DFX, a low-molecular weight and highly lipophilic drug that can be taken orally (unlike DFO). Iron chelators promote iron release from storage sites, allowing its uptake by hematopoietic tissue; further, the reduction of iron stores seems to upregulate the erythropoietin response, resulting in hemoglobin increase [4]. In various cancers, the antitumor effect of iron chelators has been reported to be due to not only their ability to bind iron but also their effects on key regulators of cellular processes such as cell cycle, apoptosis [5], cell migration, metastasis and autophagy [6]. In this context, there are still limited molecular data on DFX effects in primary MDS cells. Benarjee et al. suggested that the hematopoietic improvement observed in low-risk MDS patients treated with DFX may be mediated by inhibition of NF-κB transcriptional activity and production of inflammatory cytokines from resident immune cells in bone marrow [7]. These findings were supported by a study by Sánchez et al. showing the downregulation of NF-κB pathway in MDS patients treated with DFX [8]. In human myeloid leukemia cells, DFX suppresses mTOR signaling via enhanced expression of REDD1 and its downstream target TSC2, resulting in the inhibition of cell proliferation [9]. Furthermore, DFX was found to activate apoptosis by inhibition of Pyk2, leading to the downregulation of the Wnt/β-catenin signaling pathway in human multiple myeloma [5].

Taken together, the data indicate that DFX exerts a wide range of effects (e.g., antiproliferative and proapoptotic), and identifying the molecular mechanisms by which DFX improves outcome in MDS patients is an area of intensive investigation. Therefore, we analyzed the global gene expression of bone marrow CD34^+^ cells from untreated MDS patients and those who were given DFX and focused on the cellular pathways affected by iron chelation. Furthermore, we studied the impact of excessive iron on the gene expression regulation in untreated MDS patients with IO.

## 2. Results

### 2.1. Effect of Iron Overload on Gene Expression Regulation in MDS Patients

To address the effect of IO on gene expression regulation, comparison of patients with normal serum ferritin levels (<800 μg/L, average 326 μg/L) to those with IO (>800 μg/L, average 1864 μg/L) was performed. Of 215 differentially expressed genes (DEGs), 74 genes showed upregulation and 141 genes showed downregulation in patients with normal serum ferritin levels (*p* < 0.05; FC > 1.3). The complete list of the DEGs is available in Appendix A.

In biological process (PB) category, Gene Ontology (GO) analysis annotated the DEGs to terms (*p* < 0.05) associated with cell cycle, adaptive immune response, chromatin remodeling, positive regulation of sister chromatid cohesion, protein folding, positive regulation of interleukin-6 production, cell–cell adhesion, apoptotic process, transcription/DNA-templated, etc. (Figure 1). Pathway enrichment analysis of the DEGs identified the following Kyoto Encyclopedia of Genes and Genomes (KEGG) signaling pathways (*p* < 0.05): chronic myeloid leukemia, regulation of actin cytoskeleton, small cell lung cancer and prostate cancer (Figure 1). The list of the DEGs involved in individual GO terms/KEGG pathways and detailed parameters are available in Appendix A.

To further consider the biological significance of our results, preranked array data were analyzed by Gene Set Enrichment Analysis (GSEA) to define gene sets showing statistically significant differences between phenotypes under study. The GSEA showed that Hallmarks TNFA Signaling via NF-κB and Unfolded Protein Response are significantly enriched in iron-overloaded patients versus patients with normal ferritin levels (Figure 2). The top three Hallmark gene sets positively and negatively correlated with IO are listed in the Table 1. The complete list of the significant Hallmark gene sets (FDR < 0.25) identified by GSEA is available in Appendix A.

### 2.2. Effect of Iron Chelation on Gene Expression Regulation in MDS Patients Treated with DFX

To evaluate the impact of iron chelation on the global gene expression in MDS patients, the expression profiles of untreated patients with IO were compared to those of patients treated with DFX. Significant differential expression was detected in 88 genes; 33 genes were upregulated and 55 genes were downregulated in untreated patients with IO (*p* < 0.05, FC > 1.3). The complete list of the DEGs is available in Appendix A.

Gene Ontology analysis of the DEGs revealed a highly significant enrichment for biological processes linked with immune response (e.g., innate immune response and innate immune response in mucosa), defense response (e.g., defense response to fungus and defense response to bacterium), negative regulation of myeloid cell differentiation, cytokine production, neutrophil aggregation, positive regulation of inflammatory response and leukotriene production involved in inflammatory response (*p* < 0.01). The significantly enriched KEGG pathways were tuberculosis, cytokine-cytokine receptor interaction and Inflammatory bowel disease (*p* < 0.05) (Figure 3). The list of the DEGs involved in individual GO terms/KEGG pathways and detailed parameters are available in Appendix A.

In this dataset, Hallmark Heme Metabolism was identified by GSEA as the top gene set in patients treated with DFX versus untreated patients with IO (Figure 4). The top three Hallmark gene sets positively and negatively correlated with DFX treatment are listed in the Table 2. The complete list of the significant Hallmark gene sets (FDR < 0.25) identified by GSEA is available in Appendix A.

### 2.3. Validation of the Array Data by RT-qPCR

Real-time qPCR (RT-qPCR) was used for validating the microarray data of two highly differentially regulated genes (defensin alpha 3 (*DEFA3*) and lymphoid enhancer-binding factor-1 (*LEF1*)) in a large cohort of 82 patients (non-overloaded patients: *n* = 20; iron-overloaded patients: *n* = 28; and iron-chelated patients: *n* = 34). Significant upregulation of *DEFA3* (*p* < 0.05) was confirmed in iron-chelated patients compared to iron-overloaded patients. In agreement with the array data, *LEF1* showed significantly decreased expression (*p* < 0.01) in iron-overloaded patients compared to those without IO, and its expression increased (*p* < 0.05) in iron-chelated patients (Figure 5).

## 3. Discussion

Most MDS patients present with anemia at the diagnosis or during course of the disease and many of these patients receive regular RBC transfusions resulting in IO. Complications caused by transfusion-related IO are prevented by ICT, which may induce a hematologic improvement leading to significant reduction or complete discontinuation of blood transfusions. Several mechanisms explaining the hematologic improvement achieved by iron chelators in MDS patients have been proposed: a direct effect on the MDS clone or bone marrow microenvironment; an increase in iron release from iron stores; an alteration in NF-κB expression; an inhibition of m-TOR signaling; and reduced oxidative stress [10]. To get more insights into the molecular background of the DFX effect in MDS, we performed gene expression profiling of CD34^+^ hematopoietic progenitor cells obtained from untreated MDS patients and those who were given DFX and searched for cellular pathways affected by iron chelation.

Initially, we compared the gene expression profiles of untreated patients with normal ferritin levels to those of patients with IO to study the effect of excessive iron on gene expression regulation. Excess iron is known to be toxic due to its ability to promote ROS formation, which initiates oxidative stress and results in DNA and mitochondrial damage. In this context, the upregulation of genes related to DNA repair and responses to stress (*TAOK1*, *FEN1*, *TDP1*, *DNAJB9*, *ZNF277*, *FBXW7*, *CTSF* and *CTSH*) was detected in iron-overloaded patients. Significant upregulation in patients with IO was also observed in *ZFP91* and *BIRC3*, which are positive regulators of NF-κB pathway. Furthermore, the GSEA showed induction of TNFA Signaling via NF-κB in iron-overloaded patients compared to patients without IO (Figure 2). Oxidative stress induced by ROS has been suggested as a potential mechanism leading to NF-κB activation in MDS patients subjected to long-term transfusions [11]. Notably, DFX (not DFO or DFP) has been shown to be an efficient inhibitor of NF-κB in MDS [11].

Excess iron is linked to cell proliferation because iron is necessary for the function of many proteins involved in DNA synthesis and cell cycle. Hence, iron may function as a promoter of tumor growth since neoplastic cells require more iron for their rapid proliferation. On the other hand, oxidative stress induced via ROS generation may lead to the suppression of cell cycle and the promotion of apoptosis [12]. In this study, cell cycle and apoptosis were significantly overrepresented GO terms in the set of DEGs (Figure 1). Cell cycle-associated genes included important negative regulators, such as *TP53* and *CDKN2D*, with decreased expression in iron-overloaded patients, suggesting diminished negative regulation of cell cycle. Similarly, decreased p53 protein levels upon IO have been demonstrated in cancer cell lines and mouse model [13]. It has been suggested that heme interacts with the p53 DNA-binding domain to promote CRM1-mediated p53 export out of the nucleus and its subsequent degradation. In contrast, iron chelator treatment increases p53 expression, resulting in growth inhibition [14].

In terms of apoptosis, the GSEA showed an activation of the process in iron-overloaded patients (Appendix A), likely associated with IO-mediated oxidative stress. Because of the key role of *TP53* in cell cycle regulation, ConsensusPathDB (CPDB) database was used to disclose the significant functional interactions between *TP53* and other deregulated genes detected in iron-overloaded patients (Figure 6). This analysis identified significant interaction network between apoptotic genes and further revealed a group of seed genes connected through gene regulatory interactions to TAF1 protein, suggesting its association with the phenotype under study. This protein is a key unit of the transcription factor II D complex that is important for transcription initiation. Upon DNA damage, TAF1 inactivates p53-dependent transcription by phosphorylation of p53 [15].

The expression of *LEF1* was downregulated in patients with IO compared to those with normal ferritin levels, and its expression was significantly increased in DFX-treated patients. The transcription factor *LEF1* plays a pivotal role in lymphoid differentiation and granulopoiesis, and its decreased expression has been shown to be a negative prognostic marker in MDS [16]. The detected expression changes in *LEF1* likely reflect a worse prognosis related to excessive iron in iron-overloaded patients and an improved outcome in iron-chelated patients.

Adaptive immune response was one of the most significant GO terms in our study (Figure 1) and comprised predominantly downregulated genes (*CD79B*, *DBNL*, *PIK3CD* and *PAG1*) in iron-overloaded patients. An insufficient adaptive immune response may be associated with an increased infection risk of MDS patients with IO. Excessive iron may impair the natural resistance to infection via inhibited production of immune cytokines or impaired immune-cell functions [17]. In contrast, a significantly longer time to infection manifestation was observed in patients undergoing ICT [18].

The biological process of protein folding was overrepresented in the DEGs and included exclusively genes (*FKBP2*, *CSNK2A1*, *ERP29*, *GANAB* and *PDIA6*) with decreased expression in iron-overloaded patients. Oxidative stress induced by IO may interfere with protein folding via the disruption of disulfide bonds or the inactivation of important enzymes and signaling molecules. The accumulation of misfolded proteins in the endoplasmic reticulum (ER) leads to the unfolded protein response, resulting in attenuation of protein synthesis, increase in chaperone protein levels and activation of the ER-associated protein degradation [19]. Herein, Hallmark Unfolded Protein Response was identified by GSEA as a significant gene set enriched in iron-overloaded patients compared to those without IO (Figure 2).

As illustrated in pathway enrichment analysis (Figure 1), the genes affected by IO were significantly enriched in pathways related to cancers (chronic myeloid leukemia, small cell lung cancer and prostate cancer), supporting the hypothesis that IO is a cofactor favoring the transformation of MDS to AML [20].

In the second part of this study, we analyzed the gene expression profiles of bone marrow CD34^+^ cells obtained from DFX-treated MDS patients to study molecular mechanisms underlying the effect of DFX. The GO analysis annotated a large set of the DEGs between untreated and DFX-treated patients to terms related to the immune response (Figure 3), and all these genes showed increased expression in DFX-treated patients. In particular, the genes encoding antimicrobial peptides (AMPs) (*DEFA1*, *DEFA1B*, *DEFA3* and *CAMP*) showed markedly upregulated expression in treated patients. Human α-defensins (*DEFA1*, *DEFA1B* and *DEFA3*) are produced mainly by neutrophils; thus, they are known as human neutrophil peptides 1-4 (HNP1-4). Their secretion is an important component of innate immunity, and they are implicated in inflammatory response as well. HNP1–3 have been shown to display various inflammatory activities, such as the induction of histamine release from mast cells and chemotactic effects on human monocytes, dendritic cells and T cells [21,22,23,24]. Miles et al. demonstrated that the release of α-defensins from dying neutrophils may prevent an excessive inflammatory response leading to further damage to healthy tissue, and it is driven via inhibition of macrophage mRNA translation [25,26].

Cathelicidin (*CAMP*) represents another gene encoding AMP whose expression was upregulated in treated patients. Cathelicidin is widely expressed in various immune cells (neutrophils, monocytes, macrophages, etc.) and is highly expressed during infection, inflammation and wound healing. During inflammation, CAMP may chemoattract some inflammatory cells or change the expression of proinflammatory cytokines (IL-8 and COX-2) [27]. Another neutrophil-related gene, *CD24* (distinctly downregulated in iron-overloaded patients without ICT), showed increased expression in iron-chelated patients. It has been demonstrated that CD24 cross-ligation may accelerate neutrophil death in a caspase-dependent fashion [28]. Altogether, we detected upregulated expression of many genes functionally related to neutrophils, which are one of the first responders to inflammation [29].

Furthermore, positive regulation of inflammatory response, cytokine production and leukotriene production involved in inflammatory response were identified as significant GO terms in this dataset (Figure 3). These terms included proinflammatory genes (*S100A8*, *S100A9*, *FABP4*, *PLA2G7*, *ALOX5AP* and *ALOX5*) with increased expression levels in DFX-treated patients. Genes *S100A8* and *S100A9* encode proteins that form a heterodimer of calprotectin, which is secreted by mammalian cells (neutrophils and epithelial cells) during inflammation and it is able to sequester iron in the presence of calcium [30]. Sanchez et al. reported downregulation of NF-κB pathway in DFX-treated patients [8] and proposed a relation to reduced inflammation; however, our data suggest an ongoing process of inflammation. On the other hand, the anti-inflammatory effect of DFX may be associated with the high expression of α-defensins which may have a protective function against tissue damage. Taken together, these data indicate a significant effect of DFX on the regulation of inflammatory response, and the detailed mechanism of this effect remains to be further investigated, especially at the protein level.

The overrepresented GO terms further included term of negative regulation of myeloid cell differentiation. All genes (*MEIS1*, *HMGB3* and *ZBTB16*) in this term showed decreased expression in iron-chelated patients. Homeobox gene *MEIS* is an important factor in normal hematopoiesis that is transcribed in human CD34^+^ hematopoietic stem cells and is downregulated following differentiation [31]. Hmgb3 protein is preferentially expressed in hematopoietic stem cells, and its downregulation is an important step in myeloid and B-cell differentiation in a mouse model [32]. Gene expression of *ZBTB16* (*PLZF*) is downregulated during the differentiation of HL-60 and NB4 cells [33], and its enforced expression in myeloid cell lines results in inhibition of proliferation and differentiation [34]. Downregulation of the abovementioned negative regulators of differentiation is important for the blood cell maturation and may contribute to the hematopoietic improvement in MDS patients treated with DFX. A positive effect of DFX on hematopoiesis is further demonstrated by positive correlation of Hallmark Heme metabolism with DFX treatment (Figure 4). An activation of heme metabolism may contribute to the hemoglobin increase seen in some MDS patients after ICT [4].

## 4. Materials and Methods

### 4.1. Patients and Samples

Bone marrow samples from 47 MDS patients were obtained during routine clinical assessments at the Institute of Hematology and Transfusion, Prague. The diagnoses were made according to the 2016 WHO classification: 5 patients had MDS with ring sideroblasts (MDS-RS), 22 patients had MDS with multilineage dysplasia (MDS-MLD), 18 patients had MDS with isolated del(5q) and 2 patients had MDS with excess blasts (MDS-EB). According to the IPSS, 19 (40%) patients were low risk, 24 (51%) were intermediate-1 risk and 4 (9%) were intermediate-2 risk. The median age of the patients was 68 years (range: 28–87 years). The median number of mutations per patient was 1 (range: 0–6). Patients were divided into three groups: Group A (10 cases, serum ferritin level < 800 μg/L, no ICT), Group B (14 cases, serum ferritin level > 800 μg/L, no ICT) and Group C (23 cases, after ICT). Deferasirox (Exjade, Novartis-Pharma GmbH) was administered to patients in Group C according to guidelines for MDS patients, and the median duration of treatment was 14 months (range: 3.5–47.3 months). Paired samples were available from three patients (tested before and after ICT) and the rest of the patients were tested either before or after treatment. Patients did not receive any other therapy during the follow-up period. A summary of the patient characteristics is presented in Table 3, and detailed patient characteristics are available in Appendix A. The patient cohort used for array data validation by RT-qPCR consisted of 42 new patients and 40 patients from the original cohort. These patients were divided into the same groups as described above (Group A, 20 cases; Group B, 28 cases and Group C, 34 cases). Patient characteristics of the validation cohort are summarized in Appendix A. All subjects provided informed consent. The study was approved by the Institutional Scientific Board and the IHBT Ethic Committee (EK14/AZVCR/06/2015) and was performed in accordance with the ethical standards of the Declaration of Helsinki.

### 4.2. Cell Separation and RNA Extraction

Bone marrow CD34^+^ cells were isolated using magnetic cell separation according to the manufacturer’s protocol (Miltenyi Biotec, Bergisch Gladbach, Germany). Total RNA was isolated by acid-guanidine-phenol-chloroform method and its quality was assessed on an Agilent 2100 Bioanalyzer (Agilent Technologies, Palo Alto, CA, USA).

### 4.3. Cytogenetic Analysis

Unstimulated BM cells were cultured for 24 h in RPMI 1640 medium with 10% fetal calf serum. The chromosomal samples were prepared and evaluated as described previously [35].

### 4.4. Gene Expression Profiling

Illumina TotalPrep RNA Amplification Kit (Ambion, Thermo Fisher Scientific, Waltham, MA, USA) was used for reverse transcription of total RNA (200 ng) to generate cDNA and for in vitro transcription to generate cRNA. Biotinylated cRNA (750 ng) was hybridized on HumanHT-12 v4 Expression Bead Chips (Illumina, San Diego, CA, USA) for 16 h at 68 °C and labeled with conjugated Cy3-streptavidin. Microarrays were scanned on BeadStation 500 (Illumina), and the raw data were extracted using BeadStudio v2.0 software (Illumina).

### 4.5. Mutational Screening

The targeted next-generation sequencing (NGS) panel (TruSight Myeloid Sequencing Panel, Illumina, San Diego, CA, USA) was used to establish the mutation profile. The panel targeted 54 genes and covered the full coding sequence of 15 genes and the exonic hotspots of 39 genes associated with myeloid malignancies. Targeted sequencing was performed on Illumina MiSeq or NextSeq500 System using Illumina v3 Reagent Kit (600 cycles) and Mid OutPut v2 Kit, respectively. We identified sequence alterations with a variant allele frequency (VAF) of ≥3%.

### 4.6. Real-Time Quantitative PCR

Real-time quantitative PCR (RT-qPCR) was used to validate the microarray data of the selected genes (*LEF1* and *DEFA3*) in a large cohort of 82 MDS patients (non-overloaded patients: *n* = 20; iron-overloaded patients: *n* = 28; and iron-chelated patients: *n* = 34). RT-qPCR was performed on a StepOnePlus Real-Time PCR System (Applied Biosystems, Foster City, CA, USA) using TaqMan Expression assays (Applied Biosystems) according to the manufacturer’s recommendation. The relative gene expression was calculated based on the 2−ΔΔCT method using the *B2M* gene for normalization.

### 4.7. Data Analysis

Quality control, background subtraction and quantile normalization of the raw array data were performed in GenomeStudio v2.0 software (Illumina), and R software lumi package v4.0.2 was used for further data processing. Differential gene expression was analyzed using standard statistical functions, edgeR package and the gtools package (|FC| and their modifications. Multiple testing correction was performed using the Holm–Bonferroni method. The microarray data presented in this manuscript were deposited in the NCBI Gene Expression Omnibus (GEO) data repository and are accessible through GEO accession number GSE160727 at http://www.ncbi.nlm.nih.gov/geo/.

Gene Ontology (GO) and pathway enrichment analyses were performed using DAVID 6.8 online tool, and the significance threshold was set as *p* < 0.05. The preranked data were further analyzed by Gene Set Enrichment Analysis (GSEA) to identify gene sets with statistically significant enrichment in different phenotypes, and the significance threshold was set as FDR < 0.25. Induced network module analysis of the DEGs was performed using ConsensusPathDB (CPDB) online database by considering the genetic interactions, biochemical interactions and gene regulatory interactions with intermediate links. To increase significance of interaction network and improve intelligibility of created network, the z-score threshold of intermediate nodes was changed to 30 (default z-score is set at 20).

Statistical analyses of the RT-qPCR data were performed using GraphPad Prism 4 software (GraphPad Software, La Jolla, CA, USA). A nonparametric unpaired test (Mann–Whitney U) was used to compare expression levels between groups.

## 5. Conclusions

The study demonstrates that DFX significantly modulates the immune response at the level of gene expression. In particular, the neutrophil-related genes seem to play an important role in this modulation. Neutrophils are major effectors of acute inflammation and release cytokines, which in turn amplify the inflammatory reactions by several other cell types. In this context, positive regulation of inflammatory response and cytokine production were determined to be significantly affected by iron chelation. The expression patterns of proinflammatory genes indicate an induction of inflammation upon DFX treatment. Suppression of negative regulators of blood cell differentiation essential for cell maturation and upregulation of heme metabolism observed in DFX-treated patients may improve hematopoiesis in these patients.

In untreated patients with IO, excessive iron was associated with the altered expression of genes involved in DNA repair and apoptosis, reflecting IO-mediated oxidative stress. Further, analysis of biological processes shows that IO interferes with cell cycle, apoptosis, adaptive immunity and protein folding. The deregulation of multiple cancer pathways in iron-overloaded patients supports the concept that IO is a cofactor promoting MDS progression. Specifically, decreased expression of *TP53* found in iron-overloaded patients may contribute to the uncontrolled growth of cells, which are exposed to genomic instability via ROS generation.

## Figures and Tables

**Figure 1 pharmaceuticals-14-00041-f001:**
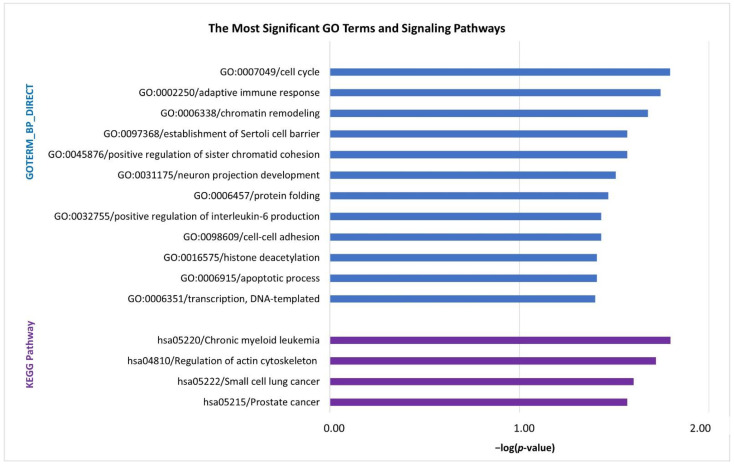
Gene Ontology (GO) biological process (BP) terms and Kyoto Encyclopedia of Genes and Genomes (KEGG) pathways (*p* < 0.05) identified by annotation and enrichment analysis of the differentially expressed genes between MDS patients with normal serum ferritin levels and those with iron overload. The bar chart represents the −log(*p*-value) of the GO term or signaling pathway.

**Figure 2 pharmaceuticals-14-00041-f002:**
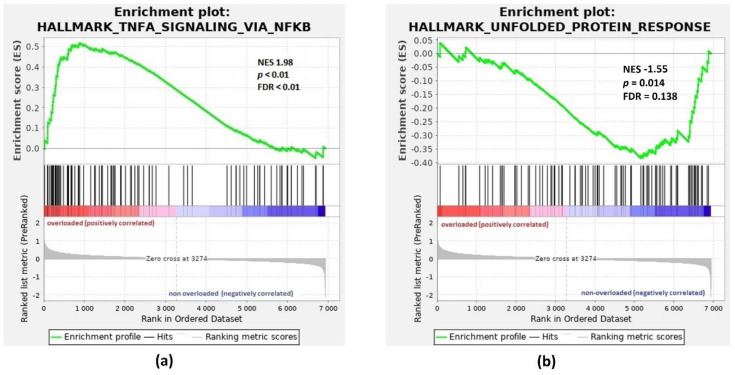
GSEA enrichment plots for Hallmark gene sets: TNFA Signaling via NF-κB and Unfolded Protein Response. Genes involved in (**a**) HALLMARK TNFA Signaling via NF-κB and (**b**) HALLMARK Unfolded Protein Response show significant enrichment in iron-overloaded patients (overloaded) versus patients with normal ferritin levels (non-overloaded). Top, the running enrichment score for the gene set as the analysis walks along the ranked list; bottom, the plot of the ranked list of all genes. *X*-axis, the rank for all genes; *Y*-axis, value of the ranking metric. ES, enrichment score; NES, normalized enrichment score; FDR, false discovery rate.

**Figure 3 pharmaceuticals-14-00041-f003:**
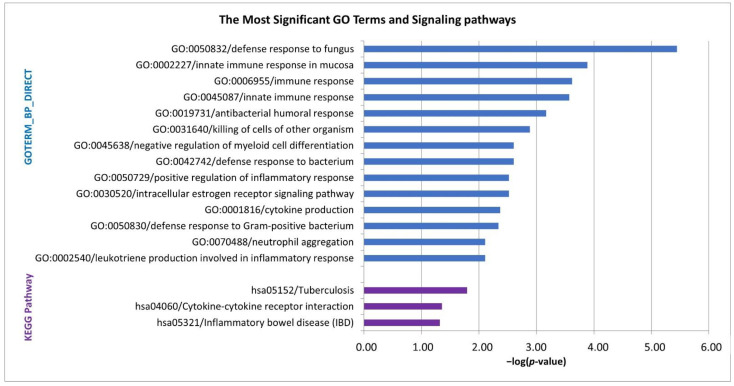
Gene Ontology (GO) biological process (BP) terms (*p* < 0.01) and Kyoto Encyclopedia of Genes and Genomes (KEGG) pathways (*p* < 0.05) identified by annotation and enrichment analysis of the differentially expressed genes between untreated MDS patients with iron overload and those treated with DFX. The bar chart represents the −log(*p*-value) of the GO term or signaling pathway.

**Figure 4 pharmaceuticals-14-00041-f004:**
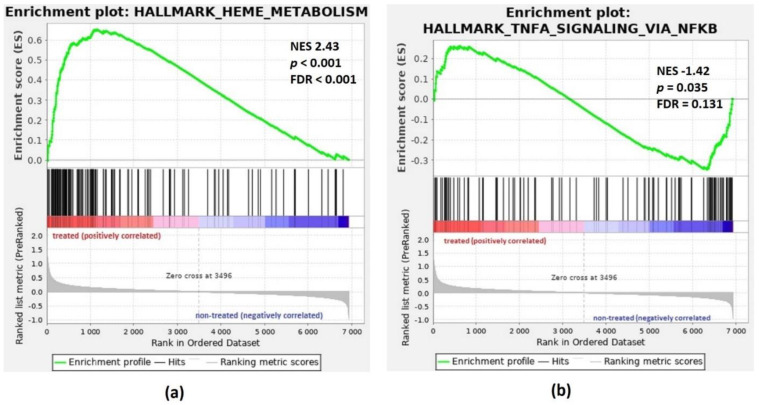
GSEA enrichment plots for Hallmark gene sets: Heme Metabolism and TNFA Signaling via NF-κB. Genes involved in (**a**) HALLMARK Heme Metabolism and (**b**) HALLMARK TNFA Signaling via NF-κB show significant enrichment in iron-chelated patients (treated) versus untreated patients with iron overload (non-treated). Top, the running enrichment score for the gene set as the analysis walks along the ranked list; bottom, the plot of the ranked list of all genes. *X*-axis, the rank for all genes; *Y*-axis, value of the ranking metric. ES, enrichment score; NES, normalized enrichment score; FDR, false discovery rate.

**Figure 5 pharmaceuticals-14-00041-f005:**
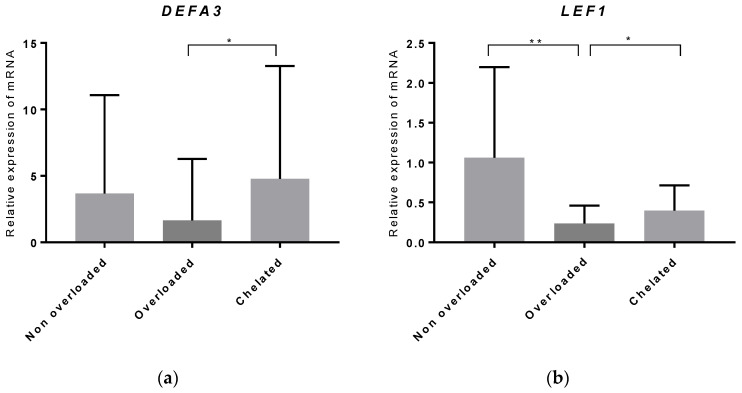
Validation of the array data by RT-qPCR. The gene expression levels of (**a**) *DEFA3* and (**b**) *LEF1* genes were determined in MDS patients with normal serum ferritin levels (non-overloaded), MDS patients with iron overload (overloaded) and MDS patients treated with DFX (chelated) by RT-qPCR. The relative expression level was normalized to the level of *B2M* expression using the 2−ΔΔCT method. Significant changes in gene expression are indicated with an asterisk (* *p* ˂ 0.05, ** *p* ˂ 0.01).

**Figure 6 pharmaceuticals-14-00041-f006:**
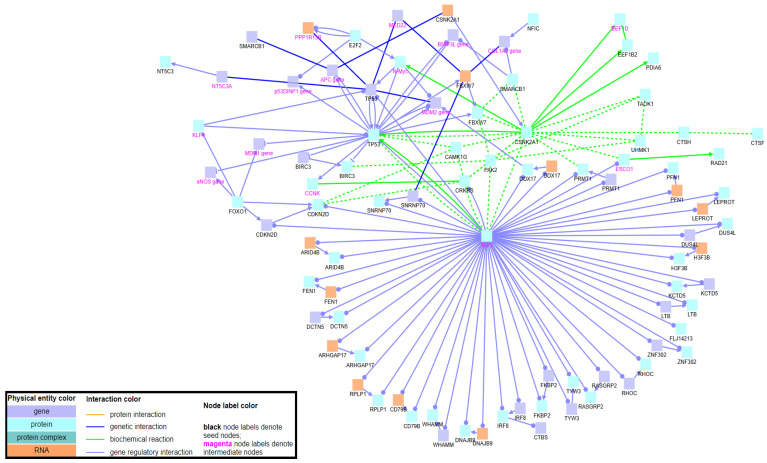
Induced network module analysis of the DEGs generated using ConsensusPathDB by considering the genetic interactions, biochemical interactions and gene regulatory interactions with intermediate genes. Nodes with black labels are seed genes, while nodes with magenta labels are intermediate nodes. Blue boxes indicate genes, light blue boxes indicate proteins and orange boxes indicate RNAs. Blue interaction lines indicate genetic interactions, green interaction lines indicate biochemical interactions and light blue interaction lines indicate gene regulatory interactions.

**Table 1 pharmaceuticals-14-00041-t001:** The top three GSEA Hallmark gene sets positively and negatively correlated with iron overload. NES, normalized enrichment score; FDR, false discovery rate.

Hallmark Gene Set	NES	*p*-Value	FDR
TNFA SIGNALING VIA NF-κB	1.98	<0.001	0.001
INTERFERON ALPHA RESPONSE	1.79	0.002	0.015
INTERFERON GAMMA RESPONSE	1.76	<0.001	0.014
PANCREAS BETA CELLS	−1.65	0.024	0.181
MITOTIC SPINDLE	−1.56	0.008	0.186
UNFOLDED PROTEIN RESPONSE	−1.55	0.014	0.138

**Table 2 pharmaceuticals-14-00041-t002:** The top three GSEA Hallmark gene sets positively and negatively correlated with iron chelation. NES, normalized enrichment score; FDR, false discovery rate.

Hallmark Gene Set	NES	*p*-Value	FDR
HEME METABOLISM	2.43	<0.001	<0.001
XENOBIOTIC METABOLISM	2.03	<0.001	<0.001
EPITHELIAL MESENCHYMAL TRANSITION	1.93	0.001	0.001
INTERFERON ALPHA RESPONSE	−1.92	<0.001	0.002
MYC TARGETS V2	−1.52	0.026	0.096
TNFA SIGNALING VIA NF-κB	−1.42	0.035	0.131

**Table 3 pharmaceuticals-14-00041-t003:** Patient characteristics.

Parameter	Group A	Group B	Group C	*p*-Value ^a^
Total, *n*	10	14	23	
Age (y), median (range)	63 (28–75)	70 (59–87)	70 (30–82)	0.12
Sex				0.74
Male (%)	4 (40)	8 (57)	13 (57)	
Female (%)	6 (60)	6 (43)	10 (43)	
Bone marrow blasts (%), median (range)	1.8 (0.2–12)	3.0 (0–8)	1.8 (0.4–6)	0.49
Hemoglobin (g/L), median (range)	82 (51–91)	89 (52–107)	84 (62–104)	0.09
Platelet count (×10^9^/L), median (range)	277 (71–766)	225 (33–1051)	158 (11–734)	0.17
ANC (×10^9^/L), median (range)	1.8 (1.0–4.3)	3.1 (0.43–12.8)	2.3 (0.4–4.9)	0.24
Diagnosis, *n*				0.54
MDS-RS	2	2	1	
MDS del(5q)	4	5	9	
MDS-MLD	3	6	13	
MDS-EB	1	1	0	
IPSS category				0.56
Low	4	7	8	
Intermediate 1	5	5	14	
Intermediate 2	1	2		
IPSS-R category				0.57
Very low/low	8	7	14	
Intermediate	1	6	8	
High	1	1	1	
Cytogenetics ^b^				0.51
Very good/good (n)	8	9	19	
Intermediate (n)	2	4	4	
Poor (n)	0	1	0	
Number of mutations, average (range)	1.3 (0–6)	1.0 (0–4)	1.1 (0–4)	0.88
Ferritin (ng/mL)	281 (162–616)	1579 (715–6889)	1309 (359–3572)	<0.00
Duration of therapy at collection (months), median (range)			14 (3.5–47.3)	

^a^*p*-values were calculated by Kruskal–Wallis test for continuous variables, Fisher’s exact test for categorical variables and log rank (Mantel–Cox) test for survival analysis. ^b^ The cytogenetic risk was based on the IPSS-R cytogenetic risk groups for MDS. Group A, MDS patients without iron overload; Group B, MDS patients with iron overload; Group C, MDS patients treated with DFX. The bold emphasizes a significant value.

## Data Availability

The data presented in this study are openly available in the National Center for Biotechnology Information (NCBI) Gene Expression Omnibus (GEO) database under accession number GSE160727 at GEO Accession viewer (nih.gov).

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
