# Peer review of "Modulation of the Immune Response by Deferasirox in Myelodysplastic Syndrome Patients"

_pharmaceuticals, 2021, doi:10.3390/ph14010041_

Round 1

Reviewer 1 Report

The study is an interesting and clinically important one in which the authors use gene expression profiling on bone marrow cells of patients with MDS, with or without iron overload and in iron overloaded patients after chelation.
The authors found iron overload affected genes of many types and that iron chelation normalized (or changed towards the normal) specific genes that were studied.
This reviewer has only a few suggestions for revisions.
1. A major weakness of the study is the use of apparently whole bone marrow cells without separation into different lineages. The authors should at least acknowledge this point.
2. I had trouble with understanding the study design. The authors are good at telling us what their conclusions are but are a bit lax about explaining which patients they study in each set of experiments. For instance the basic core studied consists of 47 patients (groups A normal ferritin, B elevated ferritin and C high ferritin on iron chelation 3 months to approximately 4 years). Reading the methods, it seems to me that each patient was studied once, therefore the authors should make this clear. For example, they say: “LEF1 showed significantly decreased expression (p<0.01) in iron-overloaded patients compared to those without IO, and its expression increased (p<0.05) after DFX therapy (Figure 5)”. The use of the word AFTER suggests that they studied individual patients at two time points and it did not seem to me that they did from what they describe. They should say “patients who were given iron chelation therapy” since we do not really know what the GEP of these patients was BEFORE chelation! Unless I am misunderstanding. In the discussion they say: “In the second part of this study, we analyzed the gene expression profiles of bone marrow cells obtained from MDS patients before and after ICT to study molecular mechanisms underlying the effect of DFX”. So DID THEY study INDIVIDUAL patients before and after ICT?
3. Furthermore there is another cohort which is mentioned: “The patient cohort assessed to validate the findings consisted of 82 patients who were divided into the same groups as described above (groups A, B and C).” What was done to these patients? Perhaps this is data that is presented in the Supplementary files….I could not figure out what is the data from these 82 patients as compared to the original 47.

Minor comment: there is an excellent review that the authors may want to cite on the effect of iron overload on malignant transformation. (This reviewer did NOT write that paper!)

Pfeihofer-Obermair, C et al. Front Oncol 2018 8:549.

Author Response

Response to Reviewer 1 Comments

  1. A major weakness of the study is the use of apparently whole bone marrow cells without separation into different lineages. The authors should at least acknowledge this point.

Response 1: In this study, gene expression analyses were not done on whole bone marrow cells but on CD34+ hematopoietic progenitor cells that are central to the pathogenesis of MDS. To emphasize the use of CD34+ cells, we added this information into the text where we use only term “bone marrow cells” (line no. 72 and 260).

  1. I had trouble with understanding the study design. The authors are good at telling us what their conclusions are but are a bit lax about explaining which patients they study in each set of experiments. For instance the basic core studied consists of 47 patients (groups A normal ferritin, B elevated ferritin and C high ferritin on iron chelation 3 months to approximately 4 years). Reading the methods, it seems to me that each patient was studied once, therefore the authors should make this clear. For example, they say: “LEF1 showed significantly decreased expression (p<0.01) in iron-overloaded patients compared to those without IO, and its expression increased (p<0.05) after DFX therapy (Figure 5)”. The use of the word AFTER suggests that they studied individual patients at two time points and it did not seem to me that they did from what they describe. They should say “patients who were given iron chelation therapy” since we do not really know what the GEP of these patients was BEFORE chelation! Unless I am misunderstanding. In the discussion they say: “In the second part of this study, we analyzed the gene expression profiles of bone marrow cells obtained from MDS patients before and after ICT to study molecular mechanisms underlying the effect of DFX”. So DID THEY study INDIVIDUAL patients before and after ICT?

Response 2: The cohort used for array-based gene expression profiling included 47 patients.  Paired samples were available from three patients (tested at two time points – before and after ICT) and the rest of the patients was tested once (either before or after treatment). For a clear understanding of the study design, we describe the study design in more detail in the section of Methods and rephrased all sentences with “before and after ICT” as suggested by reviewer.

  1. Furthermore there is another cohort which is mentioned: “The patient cohort assessed to validate the findings consisted of 82 patients who were divided into the same groups as described above (groups A, B and C).” What was done to these patients? Perhaps this is data that is presented in the Supplementary files….I could not figure out what is the data from these 82 patients as compared to the original 47.

Response 3: To demonstrate the reliability of the array data, the patient cohort was extended and tested by RT-qPCR. This cohort consisted of 42 new patients and 40 patients from the original cohort used in array experiments. The patient cohorts are described in more detail in the section of Methods and Materials. Patient characteristics of the validation cohort were summarized in Table S8 which was added into Supplementary Materials.

Minor comment: there is an excellent review that the authors may want to cite on the effect of iron overload on malignant transformation. (This reviewer did NOT write that paper!)

Pfeihofer-Obermair, C et al. Front Oncol 2018 8:549.

Response: Citation of Pfeifhofer-Obermair et al.2018 was added into the manuscript where the effect of excessive iron in relation to cancer is discussed.

Reviewer 2 Report

I would like to congratulate the authors on doing a very valuable study is a disease like MDS which does not have many treatment options and  supportive care with transfusions is a big part of management. This helps better understanding of the role of IOC in the management of MDS.

Author Response

Response to Reviewer 2 Comments

I would like to congratulate the authors on doing a very valuable study is a disease like MDS which does not have many treatment options and supportive care with transfusions is a big part of management. This helps better understanding of the role of IOC in the management of MDS.

Response: Thank you very much.

Reviewer 3 Report

While transfusion depèendency by itself is a negative prognostic factor reflecting poor bone marrow function, the ensuing transfusional iron overload has an additional dose-dependent negative impact on the survival of patients with lower risk MDS. Cardiac and hepatic dysfuntion appears to be important in this context.

In the present study, Votavova and coworkers have investigated the effects of the iron chelator deferasirox (DFX) in MDS patients, exploring the global gene expression profile before and after DFX administration at the level of CD34+ cells. Importantly, in this study patients with normal ferritin levels and with iron overload were included. Multiple cancer pathways were deregulated in iron overload patients, suggesting a role for iron overload in MDS progression. DFX treatment mainly affects biological processes related to modulation of the immune response and inflammation. Iron chelators through suppression of negative regulators of blood cell differentiation may contribute to an improvement of hematopoiesis in MDS patients.

This is an interesting paper, providing further support to the rational for the use of iron chelators in the treatment of MDS opatients with iron overload. The paper is clearly written, provides original data and is based on an appropriate methodology.

Specific comments

  • In the introduction the authors should better discuss the benefits related to the clinical use of iron chelators in MDS patients. In this context, the authgors shoud mention the results of the meta-analysis carried out by Liu et al (Clin. Exp. Med. 2020).

Author Response

Response to Reviewer 3 Comments

In the introduction the authors should better discuss the benefits related to the clinical use of iron chelators in MDS patients. In this context, the authors should mention the results of the meta-analysis carried out by Liu et al (Clin. Exp. Med. 2020).

Response: Clinical benefits of ICT in MDS are discussed in more detail in the introduction. Two new citations were used, and the findings of Liu et al. are commented.